# Comparison of Physicochemical Properties of Starches from Nine Chinese Chestnut Varieties

**DOI:** 10.3390/molecules23123248

**Published:** 2018-12-07

**Authors:** Long Zhang, Tianxiang Liu, Guanglong Hu, Ke Guo, Cunxu Wei

**Affiliations:** 1Key Laboratory of Crop Genetics and Physiology of Jiangsu Province / Key Laboratory of Plant Functional Genomics of the Ministry of Education, Yangzhou University, Yangzhou 225009, China; zhanglong@yzu.edu.cn (L.Z.); tianxiangliu1993@163.com (T.L.); 18115657147@163.com (K.G.); 2Co-Innovation Center for Modern Production Technology of Grain Crops of Jiangsu Province / Joint International Research Laboratory of Agriculture & Agri-Product Safety of the Ministry of Education, Yangzhou University, Yangzhou 225009, China; 3Institute of Forest and Pomology, Beijing Academy of Agricultural and Forestry Sciences, Beijing 100093, China; hglcau@gmail.com

**Keywords:** Chinese chestnut, starch, structural properties, thermal properties, pasting properties, digestion properties

## Abstract

Chestnut is a popular food in many countries and is also an important starch source. In previous studies, physicochemical properties of starches have been compared among different Chinese chestnut varieties growing under different conditions. In this study, nine Chinese chestnut varieties from the same farm were investigated for starch physicochemical properties to exclude the effects of growing conditions. The dry kernels had starch contents from 42.7 to 49.3%. Starches from different varieties had similar morphologies and exhibited round, oval, ellipsoidal, and polygonal shapes with a central hilum and smooth surface. Starch had bimodal size distribution and the volume-weighted mean diameter ranged from 7.2 to 8.2 μm among nine varieties. The starches had apparent amylose contents from 23.8 to 27.3% but exhibited the same C-type crystalline structure and similar relative crystallinity, ordered degree, and lamellar structure. The gelatinization onset, peak, and conclusion temperatures ranged from 60.4 to 63.9 °C, from 64.8 to 68.3 °C, and from 70.5 to 74.5 °C, respectively, among nine starches; and the peak, hot, breakdown, final, and setback viscosities ranged from 5524 to 6505 mPa s, from 3042 to 3616 mPa s, from 2205 to 2954 mPa s, from 4378 to 4942 mPa s, and from 1326 to 1788 mPa s, respectively. The rapidly digestible starch, slowly digestible starch, and resistant starch ranged from 2.6 to 3.7%, from 5.7 to 12.7%, and from 84.4 to 90.7%, respectively, for native starch, and from 79.6 to 89.5%, from 1.3 to 3.8%, and from 7.1 to 17.4%, respectively, for gelatinized starch.

## 1. Introduction

Starch is synthesized and stored as semi-crystalline granules in seeds, fruits, and some metamorphic roots and stems [1,2,3]. Starch is widely used not only as foods to provide nutrition for humans and animals, but also as raw materials for food and nonfood processing industries. Starches from different plant sources have different physicochemical properties, which determine their different applications [1,2,3]. The investigation of starch physicochemical properties is mainly focused on conventional starch sources such as cereal seeds, legume seeds, tuber crops, and some root tubers [1,2,3,4,5]. With the increasing demand for starch in food and nonfood industries, some nonconventional starch sources have been investigated in recent years [6,7,8,9,10].

Chestnuts are the fruits of *Castanea* spp. which belong to the beech family Fagaceae [11]. In 2016, the world production of chestnut reached about 2.26 million tons, and came from Asia (89.8%), Europe (6.3%), and the Americas (3.9%). The top five producers were Bolivia (about 0.08 million tons), China (about 1.88 million tons), Italy (about 0.05 million tons), Korea (about 0.6 million tons), and Turkey (about 0.06 million tons) [12]. The chestnut production in China reached about 83.1% of world production in 2016 [12]. The major cultivated species of edible chestnuts include the Chinese chestnut (*Castanea mollissima*) in China and Korea, *Castanea sativa* in Europe and South America, and *Castanea crenata* in Japan [11]. The edible kernel of chestnut contains some important functional components, such as polyphenols, vitamins, dietary fibers, minerals, and unsaturated fatty acids. The consumption of chestnut kernels has health benefits including anti-tumor, antimicrobial, and antioxidants [13,14]. Therefore, chestnuts have become an important and popular food in many countries.

The dry kernel of a chestnut contains about 50% starch and is an important starch source [15,16,17,18]. Therefore, there are many studies on chestnut starch [15,16,17,18,19,20,21,22,23,24,25,26,27,28]. The physicochemical properties of chestnut starches are mainly focused on *C. sativa* [15,17,19,20,21] and the Chinese chestnut [16,18,22,23,24,25,26,27,28]. For Chinese chestnuts, the effects of drying methods and thermal processing on structural and functional properties of starch have been reported in some literatures [22,24,25,27]. The identification and expression of starch synthesis related genes are analyzed during kernel development of Chinese chestnuts [26,28]. The physicochemical properties of starches are compared among different Chinese chestnut varieties [16,18]. For examples, Liu et al. [16] investigated the structural and functional properties of starches from four Chinese chestnut varieties and found that starches have significantly different morphologies and granule sizes, but have similar amylose contents and the same crystalline type. Hao et al. [18] investigated the physicochemical characteristics of starches from six Chinese chestnut varieties growing in different geographical conditions and found that the varieties and geographical distribution significantly influence the characteristics of chestnut starch. A large number of Chinese chestnut varieties have been cultivated in China. The different growing conditions influence the physicochemical properties of starches. Therefore, it is very important to investigate the physicochemical properties of starches from different varieties growing under the same conditions. However, such studies are little reported.

In this study, nine Chinese chestnut varieties were planted in the same farm and grew under the same conditions. Starches were isolated from fresh kernels of mature nuts, and their physicochemical properties were systematically investigated using multiple physicochemical techniques, including X-ray powder diffraction (XRD), attenuated total reflectance Fourier transform infrared (ATR-FTIR), small-angle X-ray scattering (SAXS), thermogravimetry analysis (TGA), differential scanning calorimeter (DSC), rapid visco analyzer (RVA), and some routine physicochemical measuring methods. Our objective was to investigate and compare the physicochemical properties of starches from different varieties under the same growing conditions. This study may provide some information for the quality breeding of Chinese chestnut varieties and their utilization in food and non-food industries.

## 2. Results and Discussion

### 2.1. Weight and Starch Content of Kernels

The fresh nuts of nine Chinese chestnut varieties are presented in Figure 1. They had significantly different colors and sizes. The weight of nuts and kernels were measured (Table 1). Different varieties had significantly different fresh nut weights ranging from 6.4 to 12.2 g/nut, fresh kernel weights ranging from 5.5 to 10.5 g/kernel, and dry kernel weights ranging from 2.5 to 5.6 g/kernel, with the highest for the Dahongpao and the lowest for the variety 3113. The starch content in dry kernel ranged from 42.7 to 49.3% (Table 1). The starch content in the present study is in the ranges reported by Liu et al. [16], which showed that four varieties of Chinese chestnut had starch contents from 42.4 to 53.8%. However, Hao et al. [18] reported that the starch contents ranged from 58.3 to 63.6% among six Chinese chestnut varieties. The obviously high starch contents in their report might result from the different measuring methods in starch content, variety genotype backgrounds, and plant growing conditions. In addition, the 71.4% and 78.8% starch contents have also been reported in *C. sativa* varieties [15,17]. The starch contents are influenced by plant species though they are in the same genus. The high starch content indicates that chestnuts have potential as starch sources for food and non-food applications. Therefore, the information for starch content provides a scientific basis to develop chestnut varieties.

### 2.2. Morphology and Size Distribution of Starch Granule

The Figure 2A–C shows the morphology of starch granules from Chinese chestnut variety 3113 under normal light microscope, polarized light microscope, and scanning electron microscope, and the starches from the other varieties had similar morphologies to the 3113 starch (data not shown). The nine chestnut starches all exhibited round, oval, ellipsoidal, and irregular polygonal in shapes with a central hilum and smooth surface. Similar morphologies were also reported for the chestnut starches [18,29]. The granule size was analyzed with a laser diffraction particle size analyzer. The nine chestnut starches all had bimodal size distributions with small granules from 0.5 to 1.5 μm and large granules from 1.5 to 25 μm (data not shown). The bimodal size distribution is reported in other chestnut starches [30]. The volume percentage of small granule starch ranged from 9.76% for Mi 6 to 11.84% for Mi 5 among the nine Chinese chestnut varieties (Table 2). The bimodal size distribution of large and small granules is characteristic of wheat and barley endosperm starches. The large granules form at the early stage of endosperm development, and account for more than 70% of the total starch weight but less than 10% of the granules by number. The small granules form at the middle and late stages of endosperm development, and account for over 90% of the granules by number but less than 30% of the total starch by weight [31,32]. The large starch granules have more surface pores and internal channels, higher amylose content, and lower swelling power and hydrolysis rate than the small starch granules [32,33]. Starches with predominantly small granules have potential uses as a fat replacement, fine printing paper, plastic sheet, and a carrier material in cosmetics [31]. The proportion of large and small granules differed among different chestnut varieties, which might influence the physicochemical properties of starches. The granule sizes are presented in Table 2. The nine Chinese chestnut starches had granule sizes d(0.1) from 1.7 to 2.3 μm, d(0.5) from 6.9 to 7.9 μm, d(0.9) from 12.2 to 14.0 μm, and D[4,3] from 7.2 to 8.2 μm. Similar granule size was also reported in six Chinese chestnut starches [18].

### 2.3. Iodine Absorption Spectrum and Amylose Content of Starch

The iodine absorption spectra of nine starches were determined, their derived parameters including maximum absorption wavelength, iodine blue value, and absorbance ratio of OD 620 to OD 550 (OD620/550) had no significant differences among nine starches (data not shown), but the apparent amylose contents ranged from 23.8 to 27.3% (Table 3). Some literatures report the amylose contents ranging from 21.2 to 29.8% in Chinese chestnut starches [16,18], from 21.5 to 32.8% in *C. sativa* starches [15,20,21]. The amylose is a major component of starch and plays an important role in structural and functional properties. Though the differences in measuring methods between different studies made their data hard to compare, the variations in amylose contents were detected among different varieties in the present study. The different amylose contents of nine Chinese chestnut starches might result from their different genotype backgrounds.

### 2.4. Crystalline Structure of Starch

The XRD patterns of nine Chinese chestnut starches are shown in Figure 3. According to XRD patterns, plant starches are classified into A-, B- and C-type [34]. The nine Chinese chestnut starches exhibited strong diffraction peak at about 17° 2θ and some small peaks at about 5.6°, 15°, 20°, and 23° 2θ (Figure 3). The diffraction peak at about 5.6° 2θ is a characteristic peak of B-type crystallinity, that at 23° 2θ is a characteristic peak of A-type crystallinity, and that at 20° 2θ is an amylose-lipid complex diffraction peak. The XRD patterns showed that the nine Chinese chestnut starches contained both A- and B-type crystallinity and belonged to C-type starch. Though their XRD patterns were very similar, two shoulder peaks were visible in Heishanzhai 7, Mi 5, and Mi 6 starches, indicating that the three starches contained a high proportion of B-type crystallinity and were C_B_-type starches, and the other starches had typical C-type starches (Figure 3). Some literatures report that starch from Chinese chestnut shows C-type [16], and starch from *C. sativa* exhibits B-type [19,20], and starch from *C. crenata* has C_B_-type [29]. The relative crystallinities ranged from 18.5 to 21.2% among nine Chinese chestnut starches (Table 3).

### 2.5. Short-Ranged Ordered Structure of Starch

The ATR-FTIR spectra of nine Chinese chestnut starches are shown in Figure 4. The ATR-FTIR spectrum is sensitive to the short-range ordered structure, defined as the double-helical order, as opposed to the long-range ordered structure related to the packing of double helices [35,36]. The FTIR spectrum of starch shows bands at 1150–1100 cm^−1^ (C-O, C-C, and C-O-H stretching) and 1100–900 cm^−1^ (C-O-H bending). The bands in the region 1100–900 cm^−1^ have been shown to be sensitive to changes in starch structure, especially the bands at 1045 and 1022 cm^−1^. The band at 1045 cm^−1^ is linked with order/crystalline region, and that at 1022 cm^−1^ arises as a result of absorption by stretching modes in amorphous structure [36]. The absorbance ratio of 1045/1022 cm^−1^ can reflect the ordered degree in the starch [36]. The nine Chinese chestnut starches had similar ATR-FTIR spectra in the region 1200–900 cm^−1^ (Figure 4) and their absorbance ratios of 1045/1022 cm^−1^ ranged from 0.61 to 0.65 (Table 3).

### 2.6. Lamellar Structure of Starch

The semi-crystalline growth rings formed by alternating amorphous and crystalline regions can be detected by SAXS. The SAXS patterns of nine Chinese chestnut starches are shown in Figure 5. All spectra were normalized to equal intensity at high q (0.2 Å^−1^) to account for variations in sample concentration, leading to the spectra being at the same relative scale and directly comparable. The main scattering peak at scattering vector of about 0.066 Å^−1^ is thought to arise from the periodic arrangement of alternating crystalline and amorphous lamellae of amylopectin and corresponds to lamellar repeat distance or Bragg spacing [37]. The peak position reflects the size of lamellae and may differ for starch originated from different plants, while the peak intensity reflects the electron density difference between the crystalline and amorphous regions of the lamellae. The lamellar repeat distance (D) can be calculated from the peak position (Smax) according to D = 2π/Smax [37,38]. The nine Chinese chestnut starches had lamellar peak intensity from 119 to 150 and lamellar distance from 9.27 to 9.49 nm (Table 3).

### 2.7. Thermal Properties of Starch

The thermal properties of nine Chinese chestnut starches were measured with a DSC. The DSC thermographs are presented in Figure 6, and thermal parameters are shown in Table 4. Though the gelatinization enthalpy was similar, but gelatinization temperatures had some differences among nine starches. The gelatinization onset, peak, and conclusion temperatures ranged from 60.4 to 63.9 °C, from 64.8 to 68.3 °C, and from 70.5 to 74.5 °C, respectively. Similar gelatinization temperatures are reported in starches from Chinese chestnuts cultivated in three different regions of Korea [39]. The gelatinization onset temperature (57.1 °C), peak temperature (61.9 °C), and conclusion temperature (67.9 °C) in *C. sativa* starch [20] are lower than those in Chinese chestnut starches. Liu et al. [16] reported that the gelatinization onset, peak, and conclusion temperatures range from 59.5 to 61.2 °C, from 63.7 to 65.3 °C, and from 68.5 to 70.6 °C, respectively, among four Chinese chestnut varieties, but their gelatinization enthalpies are similar. In the study of Liu et al. [16], the starch-to-water ratio is 1:2. In the present study, the starch-to-water ratio is 1:3. The different water contents influence the thermal parameters [40].

### 2.8. Pasting Properties of Starch

The pasting properties of starch are important functional properties and determine the quality and utilization of starch in food industry. The pasting properties of starch have been analyzed in some papers [41,42]. The pasting properties of nine chestnut starches were measured using an RVA. The pasting profiles and parameters are presented in Figure 7 and Table 5, respectively. The peak, hot, breakdown, final, and setback viscosities ranged from 5524 to 6505 mPa s, from 3042 to 3616 mPa s, from 2205 to 2954 mPa s, from 4378 to 4942 mPa s, and from 1326 to 1788 mPa s, respectively. Liu et al. [16] reported that the peak, hot, breakdown, final, and setback viscosities range from 2658 to 3301 cP, from 2086 to 2434 cP, from 541 to 925 cP, from 3802 to 4451 cP, and from 1685 to 2017 cp, respectively, among four Chinese chestnut varieties. Peak viscosity reflects the bind ability of starch and water, and final viscosity reflects the stability of swelling granule. Breakdown viscosity is negatively correlated to the pasting resistance of starch to heat, and setback viscosity indicates the tendency of starch paste to retrogradation [43]. The pasting properties of starches are affected by granule morphology, size, amylose content, crystalline structure, and swelling power [44]. In comparison with the results of Liu et al. [16], the chestnut starches in the present study had significantly higher peak, hot, and breakdown viscosities. Though the differences in experimental conditions between different studies make their data hard to compare, the variations in pasting properties were detected among different varieties in the present study.

### 2.9. Thermal Stability of Starch

The TGA curves are usually used to evaluate the differences in thermal stability of starch caused by structural distinctions. The TGA curves of nine starches are presented in Figure 8A. Two well-defined shifts were clearly detected. There are two crystal structures in starch, one is from (between) the ordered packing including starch molecular chains and water molecules, and the other is from the ordered packing between starch molecular chains by the interaction of hydrogen bonds. The first weight loss corresponds with the destruction of the starch-water structure, while the second is the decomposition of starch and reflects the destruction of the starch-starch structure [45]. The derivative thermogravimetric (DTG) curves of starches are presented in Figure 8B and show that the decomposition temperature occurred from 309 to 311 °C. Similar TGA and DTG curves among nine chestnut starches showed that their thermal stabilities had no difference.

### 2.10. Digestion Properties of Starch

The native and gelatinized starches were in vitro digested by both porcine pancreatic amylase (PPA) and *Aspergillus niger* amyloglucosidase (AAG) (Table 6). Starch normally contains three starch components of rapidly digestible starch (RDS), slowly digestible starch (SDS), and resistant starch (RS) according to digestion degree for within 20 min, between 20 min to 2 h, and after 2 h, respectively [46]. For native starch, the RDS, SDS, and RS ranged from 2.6 to 3.7%, from 5.7 to 12.7%, and from 84.4 to 90.7%, respectively, among nine varieties. For gelatinized starch, the RDS, SDS, and RS ranged from 79.6 to 89.5%, from 1.3 to 3.8%, and from 7.1 to 17.4%, respectively. Liu et al. [16] reported that the RDS, SDS, and RS range from 9.6 to 13.8%, from 5.5 to 20.9%, and from 67.5 to 84.9% among native starches from four Chinese chestnut varieties. In comparison with the results of Liu et al. [16], the chestnut starches in the present study had very low RDS and high RS. In the study of Liu et al. [16], the enzyme concentration is 30 U PPA and 0.4 U AAG for 1 mg starch, however, that is 0.4 U PPA and 0.4 U AAG for 1 mg starch in the present study. Therefore, the low RDS and high RS in the present study was due to the low concentration of PPA and AAG in the enzyme solution. The digestion properties of native starch are affected by granule morphology and size, starch components, and crystalline structure. The gelatinization destroys the granule morphology and crystalline structure, leading to that the gelatinized starch is degraded easily, and its digestion properties are affected mainly by starch components [47,48,49].

### 2.11. Principal Component Analysis

Though many papers report the physicochemical properties of chestnut starches, the most papers report only one chestnut variety [15,20,22,23,25,26,27,28], and few papers report two varieties [19,21] or four varieties [16,24]. Many physicochemical properties between different papers cannot be compared due to the effects of different plant growing conditions and physicochemical measuring methods on starch properties, leading to that it is difficult in analyzing the relationships among physicochemical properties and visualizing the differences and similarities among varieties in terms of various starch properties. In the present studies, nine Chinese chestnut varieties growing under the same conditions were investigated, which make it possible to analyze the interrelationships between physicochemical properties and visualize the similarities and differences among Chinese chestnut varieties. The starch physicochemical properties with normal distribution were subjected to principal component analysis (PCA), and the results are shown in Figure 9. The first, second, and third principal components (PC1, PC2, and PC3) explained 40.2, 25.3, and 21.2%, respectively, of the overall variation. The loading plot of PCA can provide the information about the interrelationships between the measured physicochemical properties of starch. The properties with curves close to each other on the plot are positively correlated while those with curves in opposite directions are significantly negatively correlated. In the present study, the loading plot of physicochemical properties showed that the lamellar peak intensity (PI) was negatively correlated to the absorbance ratio of 1045/1022 cm^−1^ in FTIR spectrum (IR) and the absorbance ratio of OD620 to OD550 in iodine absorbance spectrum (OD620/550). The gelatinization onset, peak, and conclusion temperatures (To, Tp, and Tc) were significantly positively correlated to the PI and negatively correlated to the IR and OD620/550. The pasting peak and breakdown viscosities (PV and BV) were negatively correlated to the gelatinization temperatures. The digestion of gelatinized starch was positively correlated to the starch iodine blue value (IBV), apparent amylose content (AAC), PV, and BV, and negatively correlated to the PI and gelatinization temperatures (Figure 9A). The score plot of PCA provides an overview of the similarities and differences between the starches of different varieties. The distance between the locations of any two starches can reflect the degree of the difference/similarity between them. The score plot showed that starches from different varieties of Chinese chestnut had some differences in their physicochemical properties (Figure 9B). For example, the Mi 6 starch was located at the far left of the score plot with a large negative score in PC1, while the Heishanzhai 7 (HSZ) starch had a large positive score, indicating that the two starches exhibited the greatest differences in their physicochemical properties, especially the properties whose curves in Figure 9A lied relatively close to the PC1 axis.

## 3. Materials and Methods

### 3.1. Plant Materials

Nine popular varieties (3113, Benlizi, Dahongpao, Erqingzao, Heishanzhai 7, Mi 5, Mi 6, Yangguang, and Yanshanhong) of Chinese chestnut (*Castanea mollissima* Bl.) were obtained from the Institute of Forest and Pomology, Beijing Academy of Agricultural and Forestry Sciences, China. They grew in the Shachang Village, Miyun, Beijing, China, in 2017. Their mature nuts were harvested and stored at 4 °C before use.

### 3.2. Morphology, Weight, and Starch Content of Nut and Kernel

Fresh nuts were photographed using a digital camera (IXUS 750, Canon, Tokyo, Japan). Twenty nuts were randomly chosen and weighed. Their seed coats were removed, and the obtained kernels were weighted and sliced into small pieces. The kernel pieces were dried at 110 °C for 2 h and 80 °C for 2 days, and then weighted and ground extensively. The flour was passed through 100-mesh sieve, and its starch was determined using the colorimetric method of anthrone-H_2_SO_4_ following the method of Gao et al. [9].

### 3.3. Starch Isolation

The kernels were homogenized with 0.5% (*w/v*) sodium metabisulfite aqueous solution in a home blender (JYL-C93T, Joyoung, Suzhou, Jiangsu, China). The homogenate was squeezed through four layers of cheesecloth, and filtered with 100-, 200- and 300-mesh sieves. After the starch suspension was settled overnight at 4 °C, the starch precipitate was dispersed in 0.2% (*w/v*) NaOH and centrifuged (Centrifuge 5430R, Eppendorf, Hamburg, Germany) (3000× *g*, 5 min). The starch precipitate was repeatedly treated three times using 0.2% NaOH, and then washed three times with distilled water and two times with anhydrous ethanol. Finally, the starch precipitate was dried at 40 °C, ground into powders, and passed through a 100-mesh sieve.

### 3.4. Morphology Observation and Size Distribution Analysis of Starch

The 1% (*w/v*) starch suspension in 50% (*v/v*) glycerol was observed and photographed using a polarized light microscope (BX 53, Olympus, Tokyo, Japan) under normal and polarized light. The dry starch powder was photographed using an environmental scanning electron microscope (ESEM XL-30, Philips, Eindhoven, Holland) following the method of Cai et al. [49]. The granule size distribution was analyzed using a laser diffraction instrument (Mastersizer 2000, Malvern, Worcestershire, UK) following the method of Cai et al. [50].

### 3.5. Analysis of Iodine Absorption Spectrum and Determination of Apparent Amylose Content

Starch was dissolved completely in urea dimethyl sulphoxide and treated with iodine solution following the method of Man et al. [51]. The iodine absorption spectrum was scanned using a spectrophotometer (Ultrospec 6300 pro, Amersham Bioscience, Cambridge, UK). The apparent amylose content was determined from absorbance at 620 nm.

### 3.6. XRD Analysis

The starch was humidified in a moist chamber with a saturated solution of NaCl for 1 week. The sample was analyzed using an X-ray powder diffractometer (D8, Bruker, Karlsruhe, Germany) following the method of Cai et al. [50]. The test settings were as follows: scanning from 3° to 40° 2θ, step size 0.02°, and X-ray beam at 200 mA and 40 kV.

### 3.7. ATR-FTIR Analysis

The starch was analyzed using a FTIR spectrometer (7000, Varian, Santa Clara, CA, USA) with a DTGS detector equipped with an ATR cell following the method of Cai et al. [50]. The spectrum baseline was chosen from 1200 to 800 cm^−1^, and the deconvolution was carried out using a half-width of 19 cm^−1^ and an enhancement factor of 1.9.

### 3.8. SAXS Analysis

The lamellar structure of starch was analyzed using a SAXS instrument (NanoStar, Bruker, Karlsruhe, Germany) equipped with Vantec 2000 detector and pin-hole collimation for point focus geometry. The test settings were described previously by Cai et al. [50].

### 3.9. DSC Analysis

Five milligrams of starch and 15 μL of deionized water were mixed and sealed in an aluminum pan for 2 h at room temperature. The sample was scanned using a differential scanning calorimeter (200-F3, Netzsch, Selb, Germany) from 25 to 130 °C at 10 °C/min.

### 3.10. RVA Analysis

Three grams of starch was dispersed in 25 mL deionized water. The sample was analyzed using an RVA (3D, Newport Scientific, Warriewood, NSW, Australia). The starch-water slurry was dispersed by rotating the paddle at 960 rpm for the first 10 s and then a constant speed of 160 rpm. The temperature program was set as follows: holding at 50 °C for 1 min, heating to 95 °C at 12 °C/min, maintaining at 95 °C for 2.5 min, cooling to 50 °C at 12 °C/min, and holding at 50 °C for 1.4 min.

### 3.11. TGA Analysis

The TGA of starch was carried out using a TGA system (Pyris 1, PerkinElmer, Waltham, MA, USA) following the method of Zhang et al. [52]. The sample was heated from room temperature to 800 °C at 10 °C/min. The change in sample weight against temperature was measured and analyzed.

### 3.12. Determination of Digestion Properties

The native and gelatinized starches were digested by both PPA (A3176, Sigma-Aldrich, St Louis, MO, USA) and AAG (E-AMGDF, Megazyme, Bray, Ireland) following the method of Lin et al. [53]. The gelatinized starch was prepared through heating the starch-water slurry (10 mg/mL) in a ThermoMixer C (Eppendorf, Hamburg, Germany) at 98 °C and 1000 rpm for 12 min. The native and gelatinized starches were incubated in enzyme solution (20 mM sodium phosphate buffer, pH 6.0, 6.7 mM NaCl, 0.01% NaN_3_, 2.5 mM CaCl_2_, PPA (4 U/10 mg), and AAG (4 U/10 mg)) at 37 °C using a ThermoMixer C (Eppendorf, Hamburg, Germany) with shaking at 1000 rpm. The released glucose was determined using a glucose assay kit (K-GLUC, Megazyme, Bray, Ireland).

### 3.13. Statistical Analysis

The one-way analysis of variance with post hoc contrasts by Tukey’s test and the normal distribution of all measuring data by Shapiro-Wild test were carried out using the SPSS 19.0 Statistical Software Program (IBM Company, Chicago, IL, USA). The physicochemical properties with the significance of normal distribution over 0.05 were used for principle component analysis using Minitab V. 16.0 software (IBM Company, Chicago, IL, USA).

## 4. Conclusions

Chinese chestnut kernels have high starch content and are an important starch source. Starches from different varieties growing under the same conditions showed some differences in their physicochemical properties. The lamellar peak intensity was negatively correlated to the ordered structure. The gelatinization temperatures were significantly positively correlated to the lamellar peak intensity and negatively correlated to the ordered structure. The pasting peak and breakdown viscosities were negatively correlated to the gelatinization temperatures. The digestion of gelatinized starch was positively correlated to the apparent amylose content and the pasting peak and breakdown viscosity, and negatively correlated to the lamellar peak intensity and the gelatinization temperatures. This study could provide some information for the quality breeding of Chinese chestnut varieties and their applications.

## Figures and Tables

**Figure 1 molecules-23-03248-f001:**
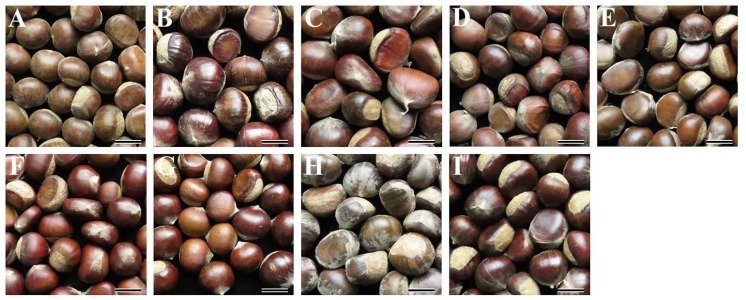
Photographs of nuts from Chinese chestnut variety 3113 (**A**), Benlizi (**B**), Dahongpao (**C**), Erqingzao (**D**), Heishanzhai 7 (**E**), Mi 5 (**F**), Mi 6 (**G**), Yangguang (**H**), and Yanshanhong (**I**). Scale bar = 20 mm.

**Figure 2 molecules-23-03248-f002:**
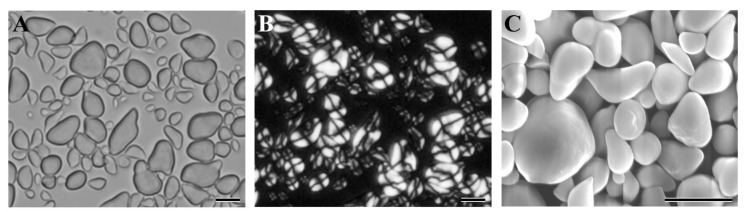
The morphology of starch granules from variety 3113 under normal light microscope (**A**), polarized light microscope (**B**), and scanning electron microscope (**C**). Scale bar = 10 μm.

**Figure 3 molecules-23-03248-f003:**
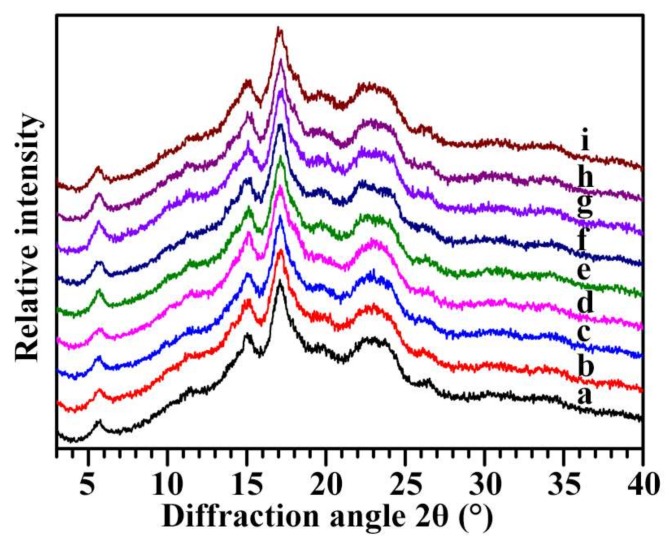
XRD patterns of starches from variety 3113 (**a**), Benlizi (**b**), Dahongpao (**c**), Erqingzao (**d**), Heishanzhai 7 (**e**), Mi 5 (**f**), Mi 6 (**g**), Yangguang (**h**), and Yanshanhong (**i**).

**Figure 4 molecules-23-03248-f004:**
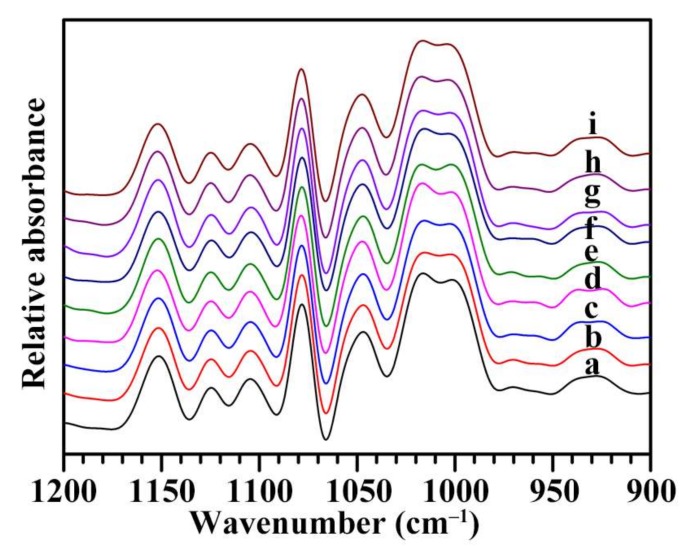
ATR-FTIR spectra of starches from variety 3113 (**a**), Benlizi (**b**), Dahongpao (**c**), Erqingzao (**d**), Heishanzhai 7 (**e**), Mi 5 (**f**), Mi 6 (**g**), Yangguang (**h**), and Yanshanhong (**i**).

**Figure 5 molecules-23-03248-f005:**
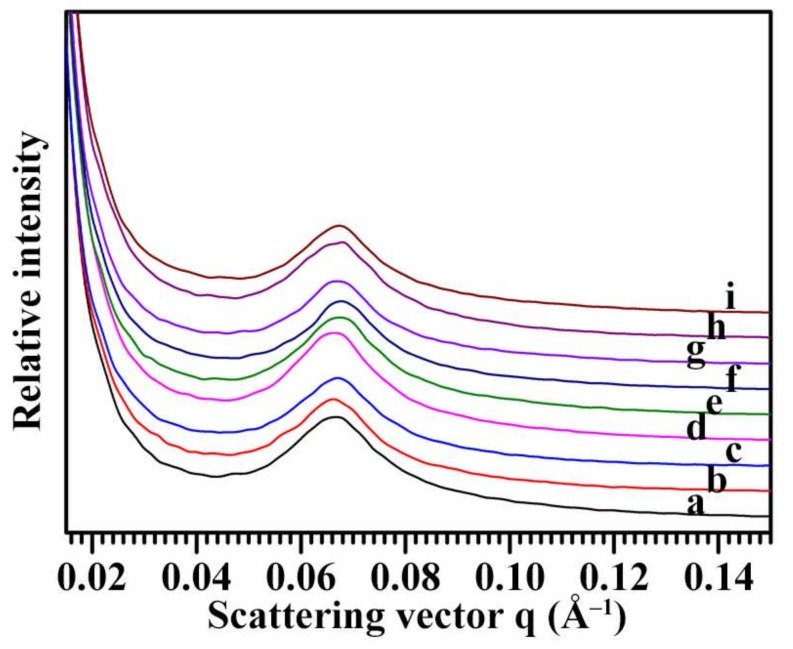
SAXS spectra of starches from variety 3113 (**a**), Benlizi (**b**), Dahongpao (**c**), Erqingzao (**d**), Heishanzhai 7 (**e**), Mi 5 (**f**), Mi 6 (**g**), Yangguang (**h**), and Yanshanhong (**i**).

**Figure 6 molecules-23-03248-f006:**
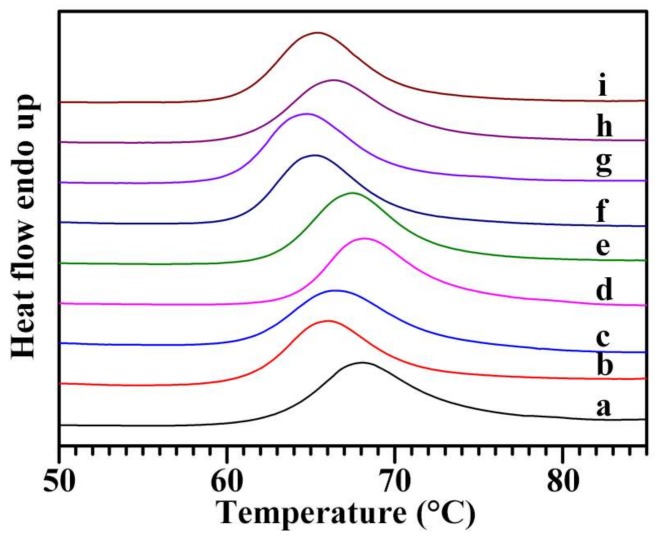
DSC thermographs of starches from variety 3113 (**a**), Benlizi (**b**), Dahongpao (**c**), Erqingzao (**d**), Heishanzhai 7 (**e**), Mi 5 (**f**), Mi 6 (**g**), Yangguang (**h**), and Yanshanhong (**i**).

**Figure 7 molecules-23-03248-f007:**
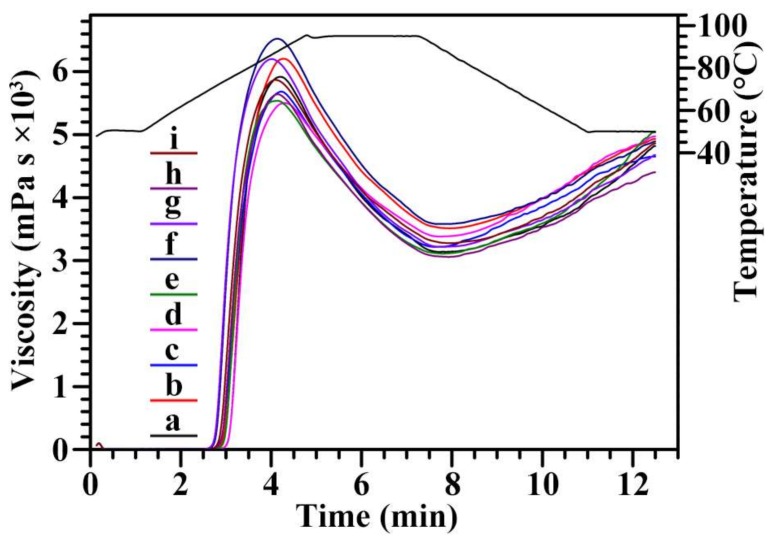
RVA profiles of starches from variety 3113 (**a**), Benlizi (**b**), Dahongpao (**c**), Erqingzao (**d**), Heishanzhai 7 (**e**), Mi 5 (**f**), Mi 6 (**g**), Yangguang (**h**), and Yanshanhong (**i**).

**Figure 8 molecules-23-03248-f008:**
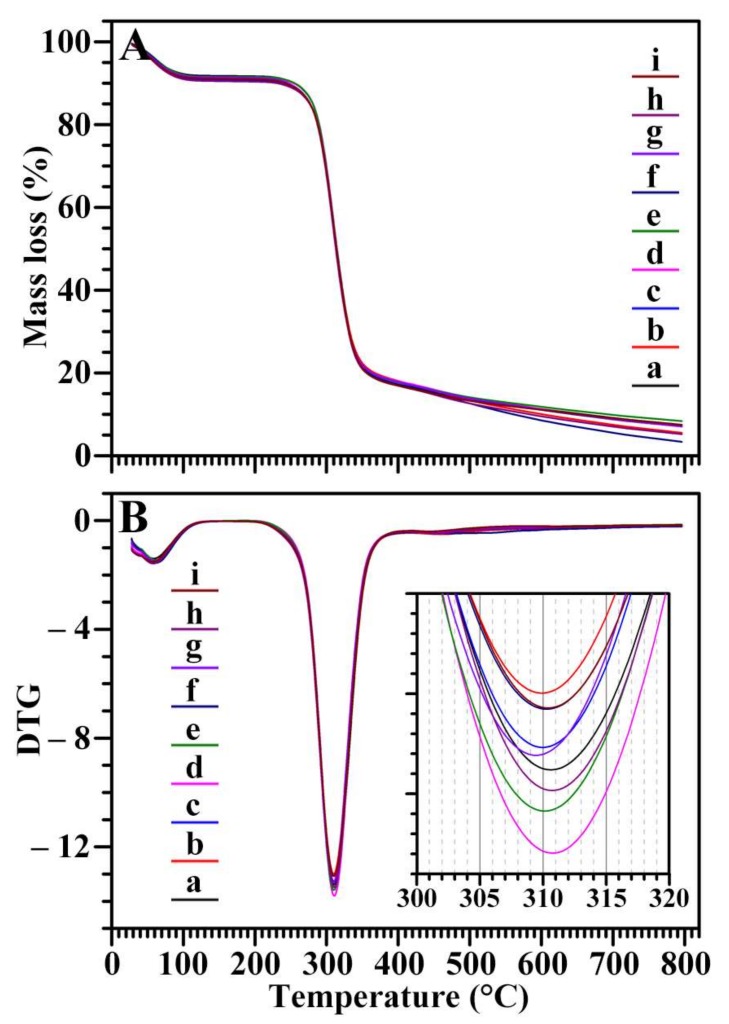
TGA curves (**A**) and DTG curves (**B**) of starches from variety 3113 (**a**), Benlizi (**b**), Dahongpao (**c**), Erqingzao (**d**), Heishanzhai 7 (**e**), Mi 5 (**f**), Mi 6 (**g**), Yangguang (**h**), and Yanshanhong (**i**).

**Figure 9 molecules-23-03248-f009:**
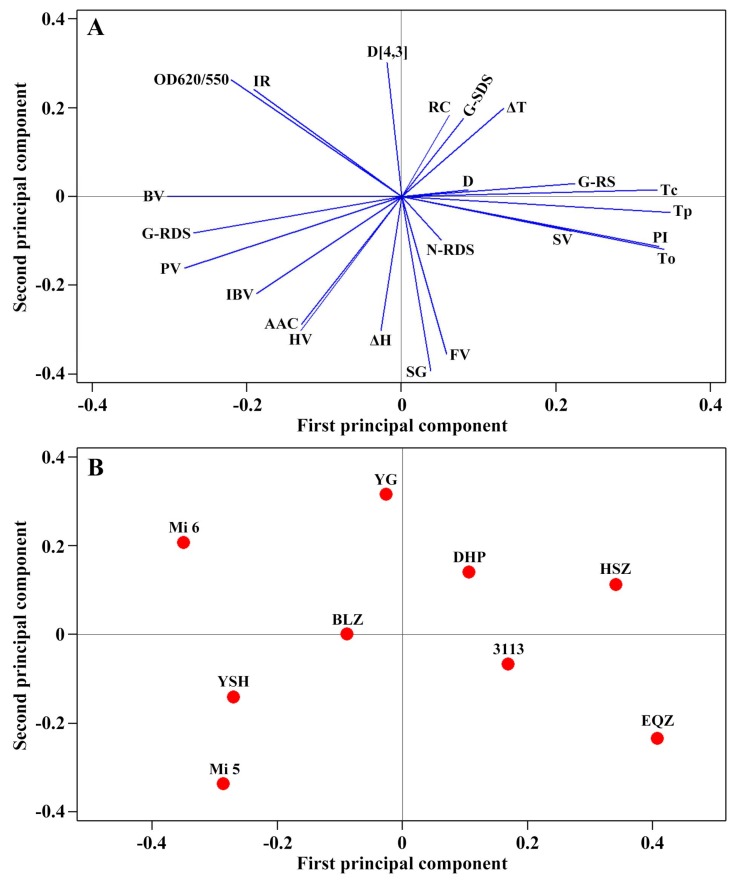
Loading (**A**) and score plots (**B**) of principal component analysis based on physicochemical properties of starches from nine Chinese chestnut varieties. The abbreviations in (**A**) are listed below: AAC, apparent amylose content; D, lamellar distance; D[4,3], volume-weighted mean diameter; IBV, iodine blue value of starch; IR, absorbance ratio of 1045/1022 cm^−1^ in FTIR spectrum; N-RDS, rapidly digestible starch in native starch; G-RDS, SDS, and RS, rapidly digestible starch, slowly digestible starch, and resistant starch in gelatinized starch; OD620/550, absorbance ratio of OD620 to OD550 in iodine absorbance spectrum of starch; PI, lamellar peak intensity; PV, HV, BV, FV, and SV, pasting peak, hot, breakdown, final, and setback viscosity, respectively; RC, relative crystallinity; SG, volume percentage of small starch granule; To, Tp, and Tc, gelatinization onset, peak, and conclusion temperature, respectively; ΔT, gelatinization temperature range; ΔH, gelatinization enthalpy. The abbreviations of varieties in (**B**) are listed below: BLZ, Benlizi; DHP, Dahongpao; EQZ, Erqingzao; HSZ; Heishanzhai 7; YG, Yangguang; YSH, Yanshanhong.

**Table 1 molecules-23-03248-t001:** Nut and kernel weights and starch content in kernel.

Varieties	Fresh Nut Weight (g/nut)	Fresh Kernel Weight (g/kernel)	Dry Kernel	
			Kernel Weight (g/kernel)	Starch Content (%)
3113	6.4 ± 0.1 ^a^	5.5 ± 0.1^a^	2.51 ± 0.03 ^a^	43.3 ± 0.2 ^a,b^
Benlizi	11.9 ± 0.5 ^e^	9.9 ± 0.4 ^d,e^	4.95 ± 0.20 ^d^	46.7 ± 1.0 ^b,c^
Dahongpao	12.2 ± 0.7 ^e^	10.5 ± 0.5 ^e^	5.61 ± 0.29 ^e^	49.2 ± 0.9 ^c^
Erqingzao	6.7 ± 0.5 ^a,b^	5.5 ± 0.4 ^a^	3.29 ± 0.22 ^b^	42.7 ± 1.0 ^a^
Heishanzhai 7	7.0 ± 0.2 ^a,b,c^	6.1 ± 0.2 ^a,b^	2.98 ± 0.11 ^b^	47.6 ± 1.5 ^c^
Mi 5	7.9 ± 0.1 ^c^	6.8 ± 0.1 ^b^	3.90 ± 0.04 ^c^	43.6 ± 1.2 ^a,b^
Mi 6	7.7 ± 0.6 ^b,c^	6.6 ± 0.5 ^b^	3.16 ± 0.22 ^b^	46.4 ± 1.0 ^a,b,c^
Yangguang	10.6 ± 0.1 ^d^	9.2 ± 0.2 ^c,d^	4.29 ± 0.08 ^c^	49.3 ± 1.2 ^c^
Yanshanhong	10.2 ± 0.3 ^d^	8.7 ± 0.3 ^c^	4.78 ± 0.17 ^d^	43.1 ± 0.6 ^a,b^
Mean ± SD	9.0 ± 2.3	7.6 ± 1.9	3.94 ± 1.04	45.8 ± 2.7
Sig.	0.160	0.205	0.802	0.137

Data are means ± standard deviations, *n =* 3. Values in the same column with different superscript letters (a–e) are significantly different (*p* < 0.05). Mean ± SD: the average value and standard deviation of nine samples. Sig.: The significance of normal distribution of nine samples by Shapiro-Wilk test.

**Table 2 molecules-23-03248-t002:** Volume percentage of small starch granules and starch granule size distribution.

Varieties	Small Granule (%)	d(0.1) (μm)	d(0.5) (μm)	d(0.9) (μm)	D[4,3] (μm)
3113	11.04 ± 0.05 ^f^	1.934 ± 0.012 ^c^	7.431 ± 0.004 ^e^	13.620 ± 0.008 ^f^	7.819 ± 0.003 ^f^
Benlizi	10.24 ± 0.04 ^b^	2.156 ± 0.015 ^f^	7.706 ± 0.001 ^g^	13.855 ± 0.004 ^h^	8.048 ± 0.001 ^g^
Dahongpao	10.70 ± 0.04 ^e^	1.988 ± 0.011 ^d^	7.291 ± 0.001 ^d^	13.073 ± 0.004 ^c^	7.607 ± 0.001 ^d^
Erqingzao	11.27 ± 0.06 ^g^	1.776 ± 0.016 ^b^	6.901 ± 0.002 ^a^	12.241 ± 0.001 ^a^	7.170 ± 0.003 ^a^
Heishanzhai 7	10.60 ± 0.03 ^d^	2.264 ± 0.013 ^g^	7.911 ± 0.001 ^i^	14.047 ± 0.003 ^i^	8.226 ± 0.001 ^i^
Mi 5	11.84 ± 0.05 ^h^	1.778 ± 0.009 ^b^	7.148 ± 0.003 ^c^	13.127 ± 0.008 ^d^	7.496 ± 0.002 ^c^
Mi 6	9.76 ± 0.03 ^a^	2.337 ± 0.014 ^h^	7.764 ± 0.001 ^h^	13.835 ± 0.002 ^g^	8.099 ± 0.001 ^h^
Yangguang	10.43 ± 0.04 ^c^	2.091 ± 0.013 ^e^	7.506 ± 0.002 ^f^	13.350 ± 0.005 ^e^	7.807 ± 0.001 ^e^
Yanshanhong	11.29 ± 0.02 ^g^	1.737 ± 0.001 ^a^	6.954 ± 0.017 ^b^	12.983 ± 0.017 ^b^	7.329 ± 0.015 ^b^
Mean ± SD	10.80 ± 0.63	2.007 ± 0.220	7.401 ± 0.358	13.348 ± 0.563	7.733 ± 0.361
Sig.	0.993	0.485	0.744	0.564	0.819

The d(0.1), d(0.5), and d(0.9) are the granule diameter at which 10%, 50%, and 90% of all the granules by volume are smaller, respectively. The D[4,3] is the volume-weighted mean diameter. Data are means ± standard deviations, *n =* 3. Values in the same column with different superscript letters (a–i) are significantly different (*p* < 0.05). Mean ± SD: the average value and standard deviation of nine samples. Sig.: The significance of normal distribution of nine samples by Shapiro-Wilk test.

**Table 3 molecules-23-03248-t003:** Apparent amylose content, relative intensity, ordered degree, and lamellar structure parameters of starch.

Varieties	AAC (%)	RC (%)	OD	Lamellar Structure Parameters	
				PI (counts)	D (nm)
3113	26.1 ± 0.5 ^b,c^	20.0 ± 0.8 ^a,b^	0.61 ± 0.02 ^a^	143 ± 8 ^a,b^	9.46 ± 0.09 ^a^
Benlizi	24.2 ± 0.7 ^a,b^	18.5 ± 0.5 ^a^	0.63 ± 0.02 ^a^	132 ± 6 ^a,b^	9.49 ± 0.00 ^a^
Dahongpao	23.8 ± 1.1 ^a^	20.7 ± 0.8 ^a,b^	0.63 ± 0.02 ^a^	136 ± 16 ^a,b^	9.38 ± 0.01 ^a^
Erqingzao	24.5 ± 0.9 ^a,b^	20.4 ± 0.5 ^a,b^	0.62 ± 0.02 ^a^	150 ± 3 ^b^	9.44 ± 0.04 ^a^
Heishanzhai 7	23.8 ± 0.9 ^a^	21.2 ± 0.9 ^b^	0.64 ± 0.02 ^a^	146 ± 7 ^a,b^	9.30 ± 0.01 ^a^
Mi 5	26.2 ± 1.1 ^b,c^	20.0 ± 1.4 ^a,b^	0.63 ± 0.01 ^a^	133 ± 8 ^a,b^	9.27 ± 0.03 ^a^
Mi 6	23.8 ± 0.9 ^a^	20.7 ± 0.4 ^a,b^	0.65 ± 0.02 ^a^	120 ± 4 ^a^	9.40 ± 0.01 ^a^
Yangguang	24.6 ± 0.6 ^a,b^	21.2 ± 0.9 ^b^	0.64 ± 0.02 ^a^	130 ± 1 ^a,b^	9.33 ± 0.14 ^a^
Yanshanhong	27.3 ± 0.5 ^c^	20.7 ± 0.6 ^a,b^	0.64 ± 0.02 ^a^	119 ± 3 ^a^	9.33 ± 0.01 ^a^
Mean ± SD	24.9 ± 1.3	20.4 ± 0.8	0.63 ± 0.01	134 ± 11	9.38 ± 0.08
Sig.	0.055	0.059	0.586	0.674	0.780

AAC: apparent amylose content; RC: relative crystallinity; OD: ordered degree; PI: lamellar peak intensity; D: lamellar distance. Data are means ± standard deviations, *n =* 3 for AAC and =2 for RC, OD, and lamellar structure parameters. Values in the same column with different superscript letters (a–c) are significantly different (*p* < 0.05). Mean ± SD: the average value and standard deviation of nine samples. Sig.: The significance of normal distribution of nine samples by Shapiro-Wilk test.

**Table 4 molecules-23-03248-t004:** Thermal parameters of starch.

Varieties	To (°C)	Tp (°C)	Tc (°C)	ΔT (°C)	ΔH (J/g)
3113	62.7 ± 0.3 ^c^	68.1 ± 0.1 ^f^	74.5 ± 0.3 ^e^	11.8 ± 0.0 ^d,e^	13.1 ± 0.1 ^a^
Benlizi	61.4 ± 0.1 ^b^	66.1 ± 0.1 ^c^	72.0 ± 0.1 ^c^	10.6 ± 0.0 ^c^	12.8 ± 0.2 ^a^
Dahongpao	61.4 ± 0.2 ^b^	66.7 ± 0.2 ^d^	73.3 ± 0.0 ^d^	12.0 ± 0.2 ^e^	13.1 ± 0.1 ^a^
Erqingzao	63.9 ± 0.1 ^d^	68.3 ± 0.1 ^f^	74.3 ± 0.0 ^e^	10.5 ± 0.1 ^b,c^	13.6 ± 0.1 ^a^
Heishanzhai 7	62.9 ± 0.1 ^c^	67.6 ± 0.1 ^e^	73.2 ± 0.2 ^d^	10.3 ± 0.1 ^a,b^	12.8 ± 0.2 ^a^
Mi 5	60.9 ± 0.0 ^b^	65.2 ± 0.0 ^b^	70.9 ± 0.0 ^a,b^	10.0 ± 0.0 ^a^	13.4 ± 0.2 ^a^
Mi 6	60.4 ± 0.0 ^a^	64.8 ± 0.1 ^a^	70.5 ± 0.1 ^a^	10.1 ± 0.1 ^a^	13.3 ± 0.4 ^a^
Yangguang	61.3 ± 0.1 ^b^	66.4 ± 0.1 ^d^	72.9 ± 0.2 ^d^	11.6 ± 0.1 ^d^	12.8 ± 0.4 ^a^
Yanshanhong	60.9 ± 0.1 ^b^	65.4 ± 0.1 ^b^	71.1 ± 0.1 ^b^	10.2 ± 0.0 ^a,b^	13.2 ± 0.1 ^a^
Mean ± SD	61.8 ± 1.1	66.5 ± 1.3	72.5 ± 1.5	10.8 ± 0.8	13.1 ± 0.3
Sig.	0.226	0.564	0.467	0.053	0.345

To, Tp, and Tc: gelatinization onset, peak, and conclusion temperature, respectively; ΔT and ΔH: gelatinization temperature range (Tc–To) and enthalpy, respectively. Data are means ± standard deviations, *n* = 3. Values in the same column with different superscript letters (a–e) are significantly different (*p* < 0.05). Mean ± SD: the average value and standard deviation of nine samples. Sig.: The significance of normal distribution of nine samples by Shapiro-Wilk test.

**Table 5 molecules-23-03248-t005:** Pasting parameters of starch.

Varieties	PV (mPa s)	HV (mPa s)	BV (mPa s)	FV (mPa s)	SV (mPa s)
3113	5866 ± 49 ^c^	3104 ± 31 ^a,b^	2762 ± 18 ^f^	4745 ± 65 ^b,c^	1641 ± 37 ^c^
Benlizi	6219 ± 38 ^d^	3487 ± 74 ^d^	2732 ± 48 ^e,f^	4954 ± 62 ^d^	1467 ± 41 ^d^
Dahongpao	5722 ± 51 ^b^	3260 ± 40 ^b,c^	2462 ± 56 ^b,c^	4650 ± 35 ^b^	1390 ± 42 ^a,b^
Erqingzao	5524 ± 21 ^a^	3319 ± 64 ^c^	2205 ± 84 ^a^	4942 ± 27 ^d^	1623 ± 50 ^c^
Heishanzhai 7	5527 ± 23 ^a^	3125 ± 59 ^a,b^	2402 ± 46 ^b^	4913 ± 35 ^d^	1788 ± 79 ^d^
Mi 5	6505 ± 41 ^e^	3616 ± 58 ^d^	2889 ± 51 ^g^	4942 ± 50 ^d^	1326 ± 27 ^a^
Mi 6	6192 ± 23 ^d^	3238 ± 23 ^b,c^	2954 ± 29 ^g^	4663 ± 32 ^b^	1425 ± 53 ^a,b^
Yangguang	5609 ± 91 ^a^	3042 ± 98 ^a^	2567 ± 22 ^c,d^	4378 ± 68 ^a^	1336 ± 33 ^a^
Yanshanhong	5860 ± 27 ^c^	3220 ± 67 ^b,c^	2640 ± 45 ^d,e^	4829 ± 34 ^c,d^	1609 ± 33 ^c^
Mean ± SD	5892 ± 345	3268 ± 185	2624 ± 241	4780 ± 192	1512 ± 160
Sig.	0.320	0.486	0.954	0.098	0.417

PV: peak viscosity; HV: hot viscosity; FV: final viscosity; BV: breakdown viscosity (PV-HV); SV: setback viscosity (FV-HV). Data are means ± standard deviations, *n* = 3. Values in the same column with different superscript letters (a–g) are significantly different (*p* < 0.05). Mean ± SD: the average value and standard deviation of nine samples. Sig.: The significance of normal distribution of nine samples by Shapiro-Wilk test.

**Table 6 molecules-23-03248-t006:** Digestion properties of starch.

Varieties	Native Starch			Gelatinized Starch		
	RDS (%)	SDS (%)	RS (%)	RDS (%)	SDS (%)	RS (%)
3113	2.6 ± 0.3 ^a^	12.6 ± 0.0 ^c^	84.8 ± 0.3 ^a^	86.5 ± 1.3 ^b^	3.5 ± 0.8 ^c,d^	10.0 ± 0.5 ^b^
Benlizi	3.7 ± 1.1 ^a^	11.3 ± 1.0 ^b,c^	85.0 ± 0.1 ^a^	84.8 ± 0.6 ^b^	3.1 ± 0.2 ^b,c,d^	12.1 ± 0.8 ^b,c^
Dahongpao	3.0 ± 0.0 ^a^	12.5 ± 0.3 ^c^	84.4 ± 0.3 ^a^	80.4 ± 0.2 ^a^	2.4 ± 1.1 ^a,b,c,d^	17.3 ± 1.3 ^d^
Erqingzao	3.3 ± 1.0 ^a^	10.0 ± 0.2 ^b^	86.7 ± 0.7 ^a^	80.7 ± 1.1 ^a^	1.9 ± 0.6 ^a,b,c^	17.4 ± 0.9 ^d^
Heishanzhai 7	3.5 ± 1.3 ^a^	5.7 ± 0.9 ^a^	90.7 ± 0.4 ^b^	79.6 ± 1.4 ^a^	3.0 ± 0.6 ^b,c,d^	17.3 ± 1.3 ^d^
Mi 5	3.4 ± 0.8 ^a^	11.7 ± 0.3 ^b,c^	84.9 ± 0.5 ^a^	84.8 ± 1.3 ^b^	1.3 ± 0.5 ^a^	13.9 ± 0.9 ^c^
Mi 6	2.9 ± 0.1 ^a^	11.9 ± 0.8 ^b,c^	85.2 ± 0.9 ^a^	86.2 ± 1.2 ^b^	1.5 ± 0.5 ^a,b^	12.3 ± 1.6 ^b,c^
Yangguang	3.0 ± 0.0 ^a^	10.7 ± 0.6 ^b,c^	86.3 ± 0.6 ^a^	84.0 ± 1.4 ^b^	3.8 ± 0.5 ^d^	12.2 ± 1.5 ^b,c^
Yanshanhong	2.9 ± 0.1 ^a^	12.7 ± 0.9 ^c^	84.4 ± 1.0 ^a^	89.5 ± 1.3 ^c^	3.4 ± 0.6 ^c,d^	7.1 ± 0.8 ^a^
Mean ± SD	3.1 ± 0.4	11.0 ± 2.2	85.8 ± 2.0	84.1 ± 3.3	2.7 ± 0.9	13.3 ± 3.6
Sig.	0.770	0.006	0.002	0.562	0.398	0.292

RDS: rapidly digestible starch; SDS: slowly digestible starch; RS: resistant starch. Data are means ± standard deviations, *n* = 3. Values in the same column with different superscript letters (a–d) are significantly different (*p* < 0.05). Mean ± SD: the average value and standard deviation of nine samples. Sig.: The significance of normal distribution of nine samples by Shapiro-Wilk test.

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
