# Peer review of "Comparison of Physicochemical Properties of Starches from Nine Chinese Chestnut Varieties"

_molecules, 2018, doi:10.3390/molecules23123248_

Round 1

Reviewer 1 Report

The manuscript studied various physicochemical properties of chestnut starch. The experiments were properly conducted and the results are interesting. However, there are some concerns of this manuscript.

1. The novelty of this manuscript is not acceptable for publication since many researches for properties of chestnut already have been reported.

2. In table 3, only AAC is required. Other parameters is not required.

3. Data of starch digestion is questionable since the RS content was quite high, but the RDS content was too low. Possibly, the author could not employ appropriate method. 4. The statistical analysis according to the storage should be included in Tables 3 and 4. The statistical analysis in Figure 1 also should be included.

Author Response

Please see the PDF file.

Reviewer 2 Report

The authors were investigated and compared the physicochemical properties of starches from different varieties with the same growing condition.

The work is interesting and can be published in the Journal of Molecules if the following issues can be addressed:

1.   In section 1, the top five Chestnut's producers should be rearranged the order. It will make the audiences feel comfortable

2.   In section 2, the starch content in dry kernel ranged from 42.7 to 49.3% which is not better than others reported. Therefore, the authors should explain more obviously.

3.   In section 2.9, the TGA curves and DTG curves were not clear. The authors should re-scale the figure to help the readers observe clearly.

4.   There are many research focus on the physicochemical properties of Chestnut. The authors should give more special information about your research. Growing in the same condition of nine chestnuts which seem basic than others report.

        5.    The writing needs to improve    

Author Response

Please see the PDF file.

Reviewer 3 Report

The content of this paper was a report of the physicochemical properties of starches from 9 Chinese chestnut varieties. It was a typical physical test report and provide some useful information. However, no novel theories or technique was found. It was only a marginal contribution.

Authors selected the one-way ANOVA with post hoc contrasts by Tukey’s test and the normal distribution of all measuring data by Shapiro-Wild test. Please indicated the assumption of the post hoc contrasts and Shapiro-Wild test and explain why authors use this method, not select other statistical technique?

Author Response

Please see the PDF file.

Round 2

Reviewer 1 Report

Although my first suggestion for this paper was “rejection for review”, the author tried to answer the suggested comments with exact manner. For this reason, I would like to recommend “accept” for this manuscript.